# Numerical and Experimental Study on NOx Reduction According to the Load in the SCR System of a Marine Boiler

Jeong-Uk Lee [1],[†] , Sung-Chul Hwang [2],[†] and Seung-Hun Han [1],*

1   Department of Mechanical System Engineering, Gyeongsang National University, Tongyeong 53064, Republic of Korea; jeonglight1017@gnu.ac.kr
2   R&D Center, GET-SCR Co., Ltd., Miryang 50404, Republic of Korea
*   Correspondence: shhan@gnu.ac.kr; Tel.: +82-55-772-9150
†   These authors contributed equally to this work.

**Abstract:** Numerical analysis and experimental studies were conducted to evaluate the performance of a selective catalytic reduction (SCR) system according to the load of a 1.5-ton marine boiler. There are post-treatment methods for reducing the exhaust gas emitted from ships, such as low-sulfur oil, scrubber, a desulfurization device to remove sulfur oxides (SOx) and particulate matter, an exhaust gas recirculation system, and SCR agents to reduce nitrogen oxides (NOx). Furthermore, there are methods of using eco-friendly natural gas fuels, such as liquefied natural gas (LNG), methanol, liquefied petroleum gas, and ammonia. In the case of LNG, SOx and particulate matter are hardly emitted, and only a small amount of NOx is emitted compared to an internal combustion engine. Therefore, SCR system technology that can remove NOx needs to be applied. As a result of this study, the boiler load increased, and the flow velocity through the outlet decreased. In addition, the NOx emissions of diesel fuel and LNG fuel were reduced by 100% to 0 ppm when the boiler load ratio was 50%. When the load ratio was 75%, the NOx emissions of diesel fuel were reduced by 77.4% to 40 ppm, and those of LNG fuel were reduced by 64.1% to 24 ppm. When the load ratio was 100%, the NOx emissions of diesel fuel were reduced by 66.1% to 60 ppm, and those of LNG fuel were reduced by 47.8% to 24 ppm. In addition, the results of the numerical analysis according to boiler load were almost identical to the experimental results. Finally, this study could design an optimal SCR system through numerical analysis, according to the important parameters of the SCR system.

**Keywords:** selective catalytic reduction; marine boiler; performance evaluation; nitrogen oxides; numerical analysis

## 1. Introduction

The Special Report on Emission Scenarios by the Intergovernmental Panel on Climate Change (IPCC) presented the prospect of accelerating climate change due to emissions in the 21st century. It predicted that if fossil fuels are used continuously at their current levels, the average global temperature will rise by 1.1–6.4 °C, and the sea level will rise by 26–60 cm by the end of the 21st century. Moreover, global greenhouse gas emissions are expected to increase by 25–90% from 2000 to 2030 [1–4]. In response to this, active research is being conducted to reduce the concentration of greenhouse gases in the atmosphere around the world [5,6]. In South Korea, various studies related to carbon dioxide ($CO_2$) emission regulations are being conducted, but they are still insufficient [7–9].

Economic exchanges between countries have been stimulated due to rapid economic growth since the Industrial Revolution, and global trade volume has been increasing as a result. In particular, the amount of trade by sea is steadily increasing; thus, the amount of sea trade shows an increasing trend toward $CO_2$ emissions due to the operation of ships [10–12].

As air pollution from ports and ships has become more serious, regulations to manage greenhouse gases are being strengthened worldwide. Implementation of the Kyoto

Protocol, which is a climate change agreement, has been attempted mainly in developed countries. The International Marine Organization (IMO) has taken the initiative in preventing marine pollution and protecting the marine environment through the 'MARPOL 73/78' agreement since 1973. In particular, the 'MARPOL Annex VI,' which came into effect in 2005, presents specific international standards for regulating the emissions of air pollutants from ships. This includes reduction standards for upper emission limits of sulfur oxides (SOx) and nitrogen oxides (NOx)—the main components of marine diesel fuel—mandatory installation of 'TIER-VI' engines for built ships, and various rules to reduce emissions of particulate matters (PMs) and carbon monoxide [13–17].

The 72nd General Assembly of the IMO Marine Environment Protection Committee held in April 2018 made a decision to reduce greenhouse gas emissions from ships by 70%, by 2050 relative to 2008 [18,19]. Consequently, it is required to reduce NOx emissions by about 80% compared to Tier I from 2016, and to reduce SOx emissions from 3.5% to 0.5% from 2020, for all sea areas [20–22].

Furthermore, various efforts are being made to monitor and reduce emissions in ports around developed countries. For example, the ports of Los Angeles and Long Beach in the United States, and the port of Oslo in Norway operate real-time air quality monitoring systems. The air pollution reduction effects are being drawn by introducing various policies, such as regulating the speed of ships in the port, applying strict emission and safety standards to all trucks that regularly enter and exit port container terminals, and providing incentives for the use of low-sulfur fuels [23,24].

There are post-treatment methods for reducing the exhaust gas emitted from ships, such as using low-sulfur oil, scrubber, and desulfurization devices to remove Sox and PMs, and an exhaust gas recirculation system and selective catalytic reduction (SCR) agent to reduce NOx. Furthermore, there are methods of using eco-friendly natural gas fuels, such as liquefied nature gas (LNG), methanol, and liquefied petroleum gas [25–28]. As for low-sulfur oil, if oil prices rise compared to the current price, the operating costs will increase significantly compared to other methods. Therefore, post-treatment methods and eco-friendly gas fuels, such as LNG, are attracting attention. In the case of LNG, SOx and PMs are hardly emitted, and only a small amount of NOx is emitted compared to an internal combustion engine. Therefore, SCR system technology that can remove NOx needs to be applied [29–31]. Most exhaust gas post-treatment devices for ships are installed in the form of retrofits and repairs. In other words, an SCR system is designed only for the target engine of the ship, and is mounted on the ship. As a result, many problems, such as a decrease in the nitrogen reduction rate, a delay in reaction time, and an ammonia slip phenomenon, have occurred. Therefore, it will be possible to optimally design and install the SCR system in an existing ship if the reaction rate and nitrogen reduction rate are identified through numerical analysis, taking into account the major variables of the target engine, exhaust gas line, and flow uniformity. This study compared NOx reduction rates through numerical and experimental analyses of an SCR system according to the load of a 1.5-ton marine boiler using diesel fuel and LNG fuel.

## 2. Methodology

### 2.1. Experimental Apparatus

Figure 1 shows the 1.5-ton marine boiler and SCR reactor used in this study, and Table 1 lists the specifications of the 1.5-ton marine boiler. The design pressure is 10 kg/cm$^2$, the maximum temperature is 300 °C, and the maximum flow rate under 100% boiler load is 1500 Am$^3$/h. The converted evaporation is 1791 kg/h, the total outbreak heat cap is 965,550 kcal/h, and the boiler efficiency is 88%. In addition, the heating surface area is 35 m$^2$, the fuel consumption for diesel is 106.5 kg/h, and the fuel consumption for LNG is 110.3 kg/h.

Figure 2 shows the sectional view of the SCR system of the marine boiler. It was applied to ships as well as offshore and industrial plants, and it was possible to achieve up

to 98% NOx reduction by design. It was made light and compact for installation on a ship, and complies with Tier 3 regulations that came into effect in 2016.

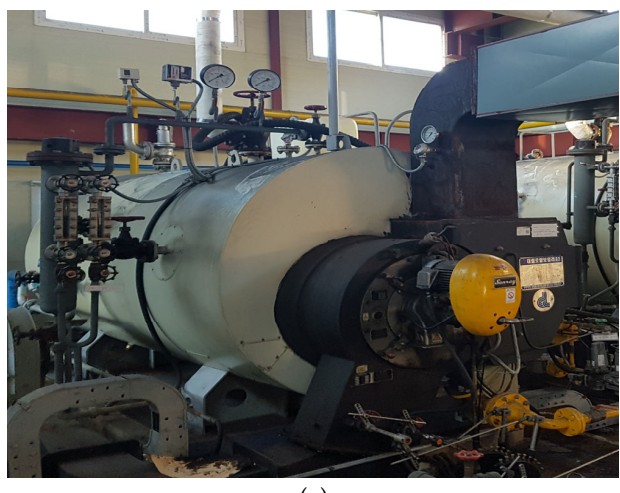

(**a**)

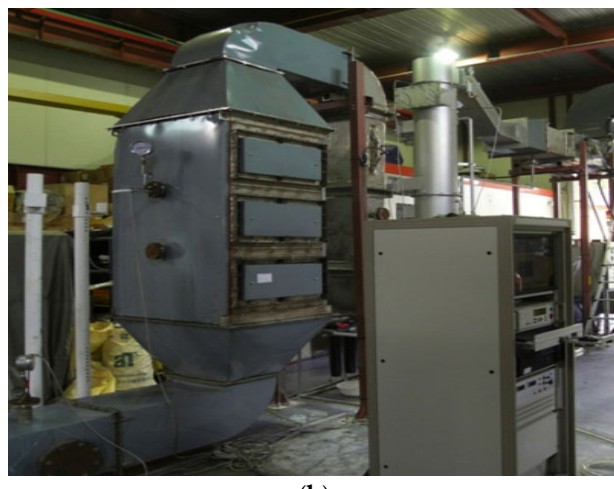

(**b**)

**Figure 1.** Photo of a 1.5-ton marine boiler using SCR systems: (**a**) 1.5-ton marine boiler; (**b**) SCR reactor.

**Table 1.** Experimental specifications in a 1.5-ton marine boiler.

| Test Condition | Value |
|---|---|
| Design pressure (kg/cm$^2$) | 10 |
| Max. temperature (°C) | 300 |
| Maximum flow rate (Am$^3$/h) | 1500 |
| Convert evaporation (kg/h) | 1791 |
| Total outbreak heat cap (kcal/h) | 965,550 |
| Efficiency (%) | 88 |
| Heating surface (m$^2$) | 35 |
| Fuel consumption, Diesel (kg/h) | 106.5 |
| Fuel consumption, LNG (kg/h) | 110.3 |

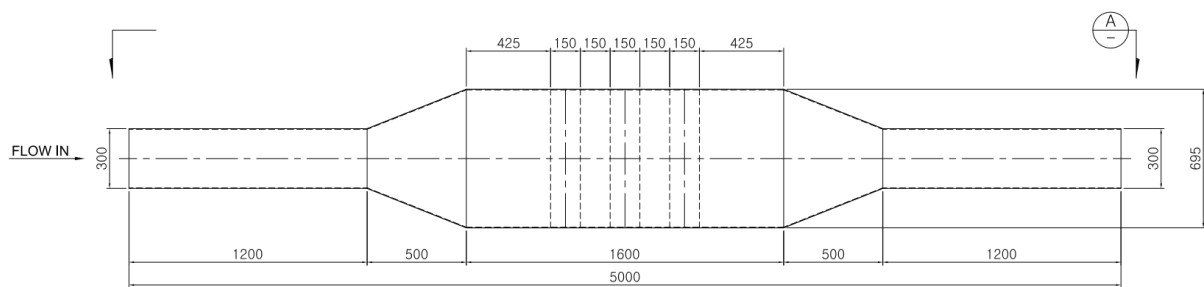

**Figure 2.** Sectional view of the SCR system.

Figure 3 shows a schematic diagram of the experimental system. It was constructed by connecting the SCR system and the post-treatment device in a 1.5-ton marine boiler. The exhaust gas flows through the flue line of the boiler and passes through the urea sup-ply system and reactor, where a selective catalytic reaction occurs to reduce NOx in the exhaust gas. The urea solution was used for the injection of urea, and the injection point was located 0.5 m away from the SCR reactor. The urea injector was of the cone angle type, and injected urea at a flow rate of 0.00033 kg/s. The NOx emission measuring devices were installed at the inlet and outlet of the flue line to measure and store the NOx reduction rates before and after the SCR system, respectively.

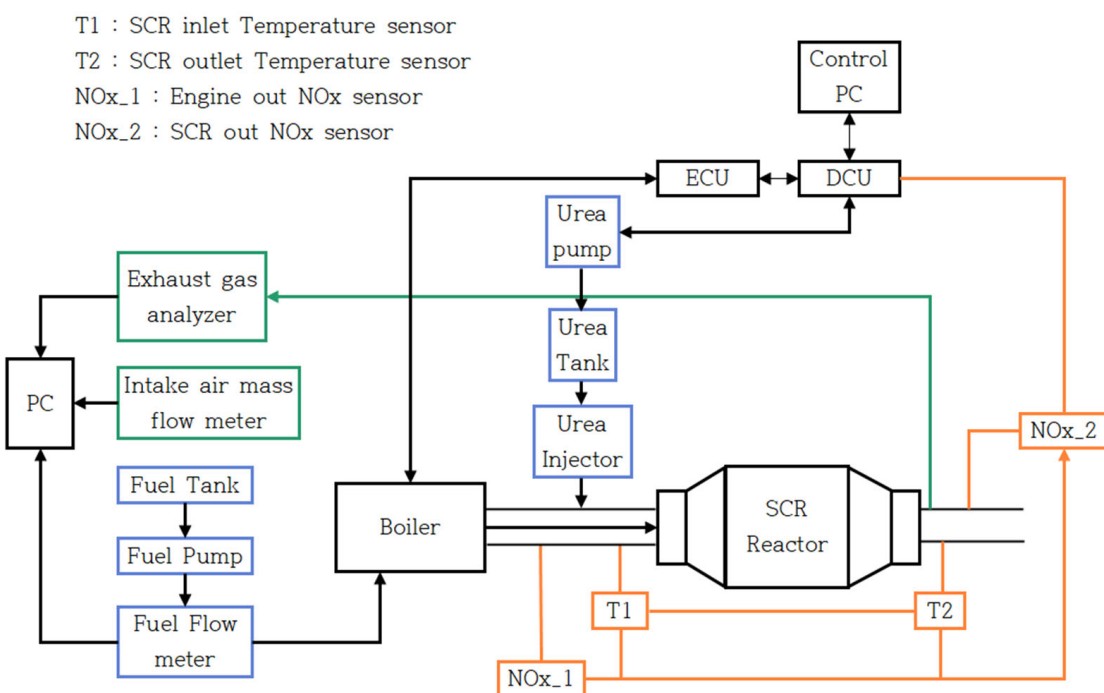

**Figure 3.** Schematic diagram of the experimental system.

In this study, the role of the gas analyzer was important to obtain accurate results. The gas analyzer used to measure NOx in the experiment was a self-developed analyzer that has been certified for performance. This analyzer was a hybrid in situ type, and compared to the existing in situ or sampling type, there was no error caused by dust or vibration, and its reliability was certified. The electronic control unit (ECU) and domain control unit (DCU) were connected to enable automatic control and real-time monitoring. Through this, a system was configured to integrate individual sensors to process and control signals.

*2.2. Numerical Analysis*

For the numerical analysis, a method previously studied by the author was used, and the boundary conditions were modified in the same way as in the experiment [32]. The results obtained through numerical analysis were compared with the experimental results to confirm the validity of the numerical analysis, and performed under the same conditions as the experiment; this was carried out in order to develop the SCR system based on the numerical analysis. The focus was on analyzing the results according to the characteristics of diesel and LNG fuels, and the results of the 1.5-ton marine boiler load.

The numerical analysis of the 1.5-ton marine boiler was conducted using commercial CFD (Computational Fluid Dynamics) code, i.e., Ansys Fluent 2020 R2, and Table 2 shows the characteristics of diesel fuel and LNG fuel by the load of a 1.5-ton marine boiler for numerical analysis. The analysis conditions were established through basic experimental research of the actual boiler SCR system.

The experimental results of the maximum flow values measured at the inlet of the flue line were 750 Am$^3$/h, 1,125 Am$^3$/h, and 1500 Am$^3$/h at 50%, 75%, and 100% boiler loads, respectively. The inlet flow velocities were 3 m/s, 4.5 m/s, and 6 m/s, respectively, the reactor porosity was 0.7, and the NOx emissions were 177 ppm for diesel fuel and 67 ppm for LNG fuel. The urea injection rates of diesel fuel and LNG fuel were set to 300 g/h and 160 g/h, respectively.

In order to analyze the reaction of the SCR system and the urea decomposition reaction, the surrounding gas flow field, which has a distinct continuum character, was analyzed from the Euler perspective. In addition, the sprayed particles of the urea solution, having

the behavior characteristics of discontinuous particles, were numerically analyzed from the Lagrangian point of view.

**Table 2.** Specifications of the diesel fuel and LNG fuel in a 1.5-ton marine boiler for numerical analysis.

| Item | | Diesel | LNG |
|---|---|---|---|
| Inlet flow velocity (m/s) | 50% load | 3.0 | |
| | 75% load | 4.5 | |
| | 100% load | 6.0 | |
| CHO content (%) | C | 86 | 76 |
| | H | 14 | 24 |
| | O | 0 | 0 |
| Porosity | | 0.7 | 0.7 |
| Urea (g/h) | | 300 | 160 |
| IN NOx (ppm) | | 177 | 67 |
| Density (kg/m$^3$) | | 831 | 415 |
| Specific gravity | | - | 0.55 |
| Cetane number | | 40–55 | - |
| Boiling point (°C) | | 180-370 | −162 |
| Kinematic Viscosity (mm$^2$/s) | | 3 | - |
| Ignition point (°C) | | 250 | 537 |
| Exhaust Inlet Velocity (m/s) | | 3–6 | |
| Exhaust Temperature (°C) | | 200 | |
| Catalyst Porosity | | 0.7 | |
| Wall | | No-slip | |
| Turbulence Model | | K–$\varepsilon$ | |

The grids of the SCR system modeling for numerical analysis were linear cell grids used for the precise design of the SCR system. In the SCR system model, the number of nodes, including the inlet, outlet, and heated wall boundary was set to 84,681, and the number of grids was set to 80,373.

The physical properties were set to values that were suitable for diesel and LNG fuels. The diesel fuel had a density of 831 kg/m$^3$; a specific gravity of 0; a cetane number of 40–55; a boiling point of 180–370 °C; a CHO content of 86% for carbon, 14% for hydrogen, and 0% for oxygen; a kinematic viscosity of 3 mm$^2$/s; and an ignition point of 250 °C. The LNG fuel had a density of 415 kg/m$^3$; a specific gravity of 0.55; a cetane number of 0; a boiling point of −162 °C; a CHO content of 76% for carbon, 24% for hydrogen and 0% for oxygen; a kinematic viscosity of 0; and an ignition point of 537 °C.

The boundary condition was set to the same state as the experiment of the SCR system. Regarding the SCR system for each boiler load, the velocity at which exhaust gas was introduced was set to 3–6 m/s, and 200 °C was applied as the temperature of the boiler exhaust gas. The no-slip condition was used as the adhesion condition of the wall, and the turbulence model was assumed to be a steady-state turbulent flow. In addition, a K–$\varepsilon$ model with verified engineering feasibility was used.

## 3. Results

### 3.1. Diesel Fuel

Figure 4 shows the numerical analysis results at a boiler load of 50%. Figure 4a shows the flow velocity of the SCR system, with an inlet velocity of 2.5–3 m/s and an outlet velocity of 10 m/s. The result of the outlet velocity being higher than the inlet velocity as it passed through the SCR reactor was the same in all numerical analyses. This result allowed the inlet velocity to be kept constant, and was considered to increase the flow velocity because the flow was discharged to the atmosphere after the SCR reactor. Figure 4b shows the pressure difference between the inlet and outlet of the reactor. As a result of checking the pressure change, the pressure difference between the inlet and the outlet was 28 Pa when the boiler load was 50%, but the outlet pressure decreased to 0 Pa when the flow



passed through the reactor. Figure 4c shows the NOx reduction rate per hour according to the boiler load. The results of the numerical analysis confirmed that NOx was introduced into the inlet at 177 ppm when the boiler load was 50%, but the NOx discharged through the outlet was reduced to 0 ppm. Furthermore, it was found that the catalytic reaction was most activated, and NOx emissions were reduced near the 2/5 point of the SCR system reactor inlet.

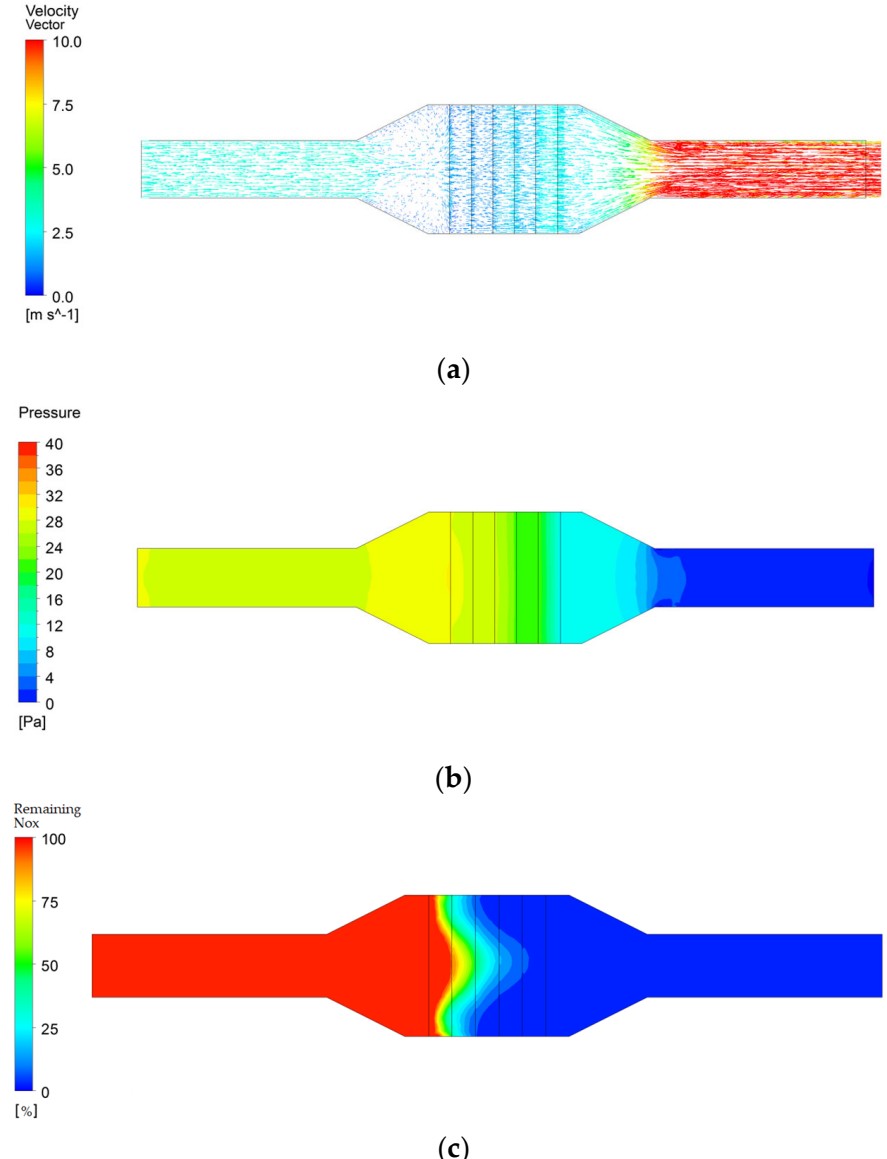

**Figure 4.** NOx reduction in the SCR system in a 1.5-ton boiler (diesel, 50% load): (**a**) velocity results for a boiler load of 50%; (**b**) pressure results for a boiler load of 50%; (**c**) NOx reduction results for a boiler load of 50%.

Figure 5 shows the numerical analysis results at a boiler load of 75%. Figure 5a shows the flow velocity of the SCR system, with an inlet velocity of 2.5–3 m/s and an outlet velocity of 10 m/s. Figure 5b shows the pressure difference between the inlet and outlet of the reactor. As a result of checking the pressure change, the inlet pressure was 28 Pa when the boiler load was 75%, but the outlet pressure decreased to 0 Pa when the flow passed through the reactor. Figure 5c shows the NOx reduction rate per hour according to the boiler load. The results of the numerical analysis confirmed that when the boiler load was 75%, NOx was introduced into the inlet at 177 ppm, but the NOx discharged through

the outlet was reduced to 40 ppm. The catalytic reaction was the most activated, and NOx emissions were reduced near the 3/5 point of the inlet of the SCR system reactor.

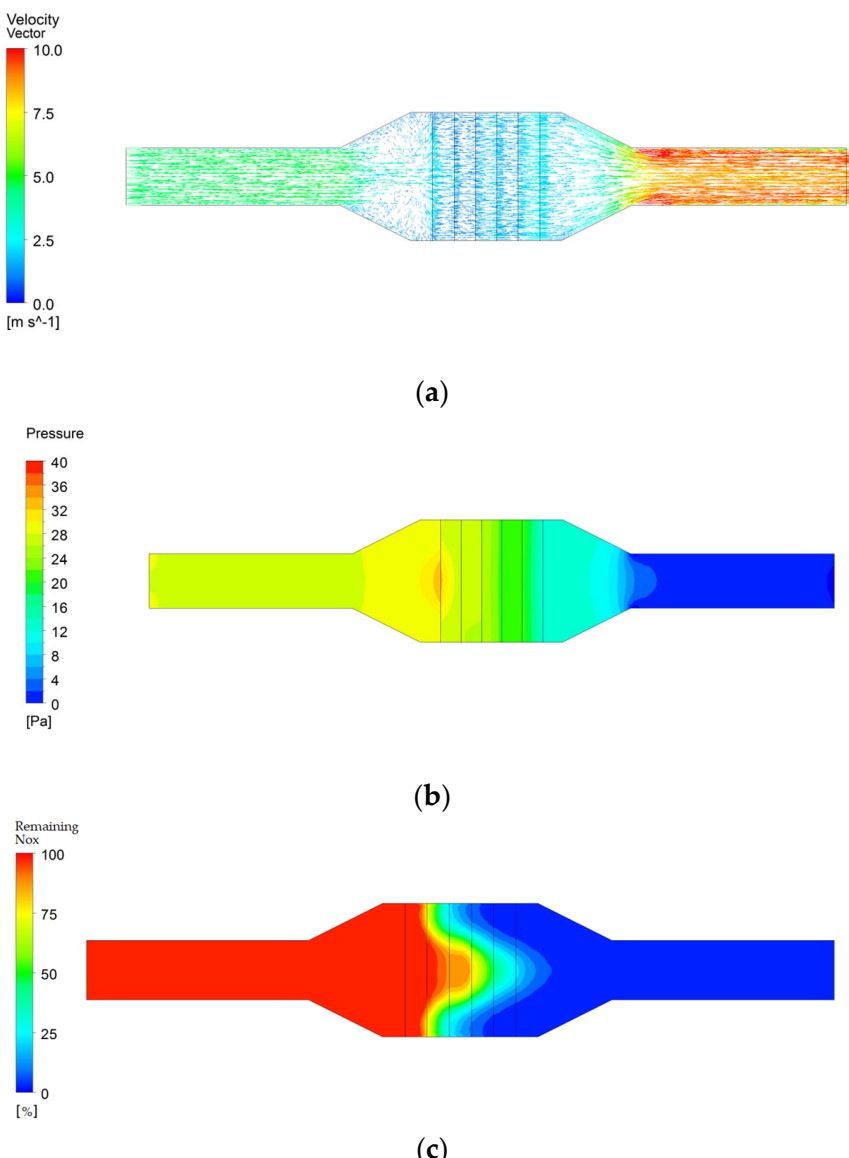

(a)

(b)

(c)

**Figure 5.** NOx reduction in the SCR system in a 1.5-ton boiler (diesel, 75% load): (**a**) velocity results for a boiler load of 75%; (**b**) pressure results for a boiler load of 75%; (**c**) NOx reduction results for a boiler load of 75%.

Figure 6 shows the numerical analysis results at a boiler load of 100%. Figure 6a shows the flow velocity of the SCR system with an inlet velocity of 2.5–3 m/s and an outlet velocity of 10 m/s. Figure 6b shows the pressure difference between the inlet and outlet of the reactor. As a result of checking the pressure change, the pressure difference between the inlet and the outlet was 28 Pa when the boiler load was 100%, but the outlet pressure decreased to 0 Pa when the flow passed through the reactor. Figure 6c shows the NOx reduction rate per hour according to the boiler load. The results of the numerical analysis confirmed that when the boiler load was 100%, NOx was introduced into the inlet at 177 ppm, but the NOx discharged through the outlet decreased to 60 ppm. It was found that the catalytic reaction of the reactor was the most activated, and NOx emissions were reduced at the 3/5 point of the inlet of the SCR system.

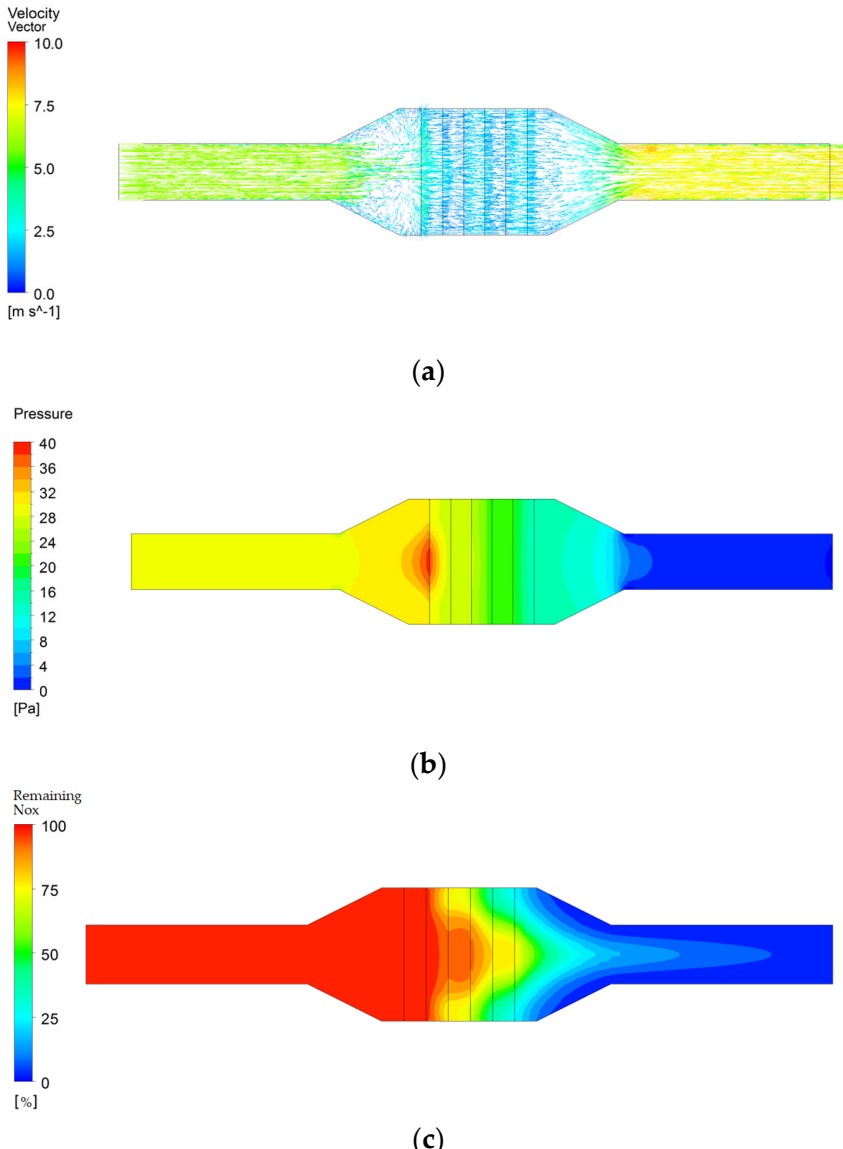

**Figure 6.** NOx reduction in the SCR system in a 1.5-ton boiler (diesel, 100% load): (**a**) velocity results for a boiler load of 100%; (**b**) pressure results for a boiler load of 100%; (**c**) NOx reduction results for a boiler load of 100%.

Figure 7 shows the NOx reduction rate through numerical analysis and experiments with the boiler SCR system when 300 g/h of urea was injected under boiler loads of 50%, 75%, and 100%. In this plot, the time was the value measured from the point at which the SCR system began operating by measuring the NOx value in the exhaust gas analyzer. After starting the measurement, the DCU calculated the appropriate urea injection amount, and urea was injected. The long resident time was due to the slow flow velocity of 3.0 to 6.0 m/s, due to the characteristics of the 1.5-ton marine boiler. When the boiler load was 50%, NOx at the inlet was detected as 177 ppm. The NOx coming out of the outlet was 0 ppm, indicating a 100% reduction. When the boiler load was 75%, NOx coming out of the outlet was 40 ppm, which corresponded to a 77.4% reduction. When the boiler load was 100%, the NOx coming out of the outlet was 60 ppm, representing a reduction of 66.1%. Furthermore, a difference in the catalytic reaction time between the numerical analysis and experimental results occurred at low boiler loads of 50% and 75%. This was considered to be a difference caused by a decrease in the activity of the catalyst when the load was low. The activity of the catalyst was lowered by operating conditions or chemical factors.

In particular, when the gas temperature was lowered, the NH3 adsorption performance was lowered. Since perfect insulation was not possible in the experimental conditions for the 50% and 75% loads, it was difficult to have ideal conditions as in the numerical analysis setting because the temperature of the gas was lowered due to the low external air temperature. In order to overcome this, we plan to conduct a study to analyze the results by installing a heater to heat the gas in the future. When the load was 100%, the numerical analysis and experimental results were almost identical, with no difference in the catalytic reaction time.

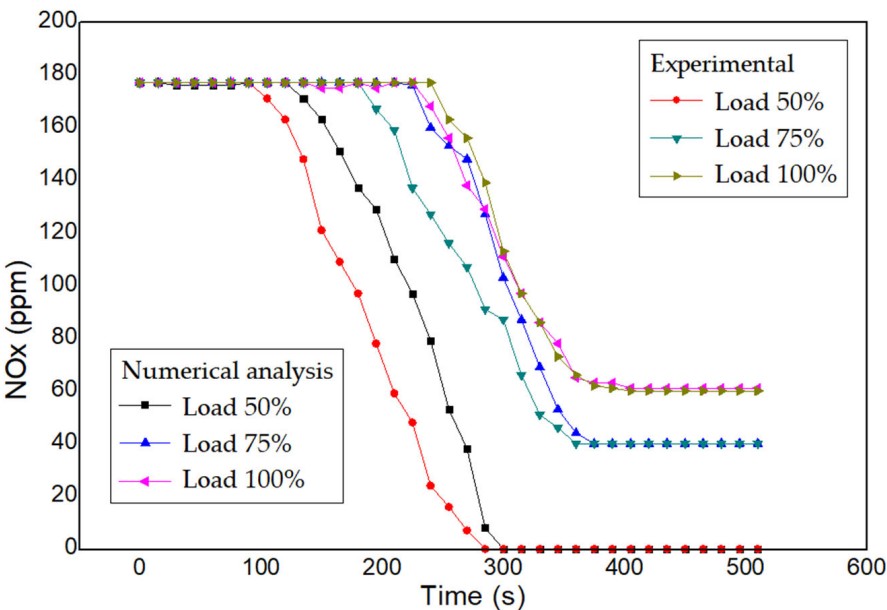

**Figure 7.** NOx emissions reduction according to the load when using diesel fuel.

*3.2. LNG Fuel*

Figure 8 shows the numerical analysis results at a boiler load of 50%. Figure 8a shows the flow velocity of the SCR system with an inlet velocity of 2.5–3 m/s and an outlet velocity of 10 m/s. Figure 8b shows the pressure difference between the inlet and outlet of the reactor. As a result of checking the pressure change, the pressure difference between the inlet and outlet was 32 Pa when the boiler load was 100%, but the outlet pressure decreased to 0 Pa when the flow passed through the reactor. Figure 8c shows the NOx reduction rate per hour according to the boiler load. The results of the numerical analysis confirmed that NOx was introduced into the inlet at 67 ppm when the boiler load was 50%, but the NOx discharged through the outlet was reduced to 0 ppm. The catalytic reaction was the most activated, and NOx emissions were reduced near the 2/5 point of the SCR system reactor inlet.

Figure 9 shows the numerical analysis results at a boiler load of 75%. Figure 9a shows the flow velocity of the SCR system with an inlet velocity of 2.5–3 m/s and an outlet velocity of 7.5–10 m/s. Figure 9b shows the pressure difference between the inlet and outlet of the reactor. As a result of checking the pressure change, the pressure difference between the inlet and outlet was 32 Pa when the boiler load was 100%, but the outlet pressure decreased to 0 Pa when the flow passed through the reactor. Figure 9c shows the NOx reduction rate per hour according to the boiler load. The results of the numerical analysis confirmed that NOx was introduced into the inlet at 67 ppm when the boiler load was 75%, but the NOx discharged through the outlet was reduced to 24 ppm. The catalytic reaction was the most activated, and NOx emissions were reduced near the 3/5 point of the inlet of the SCR system reactor.

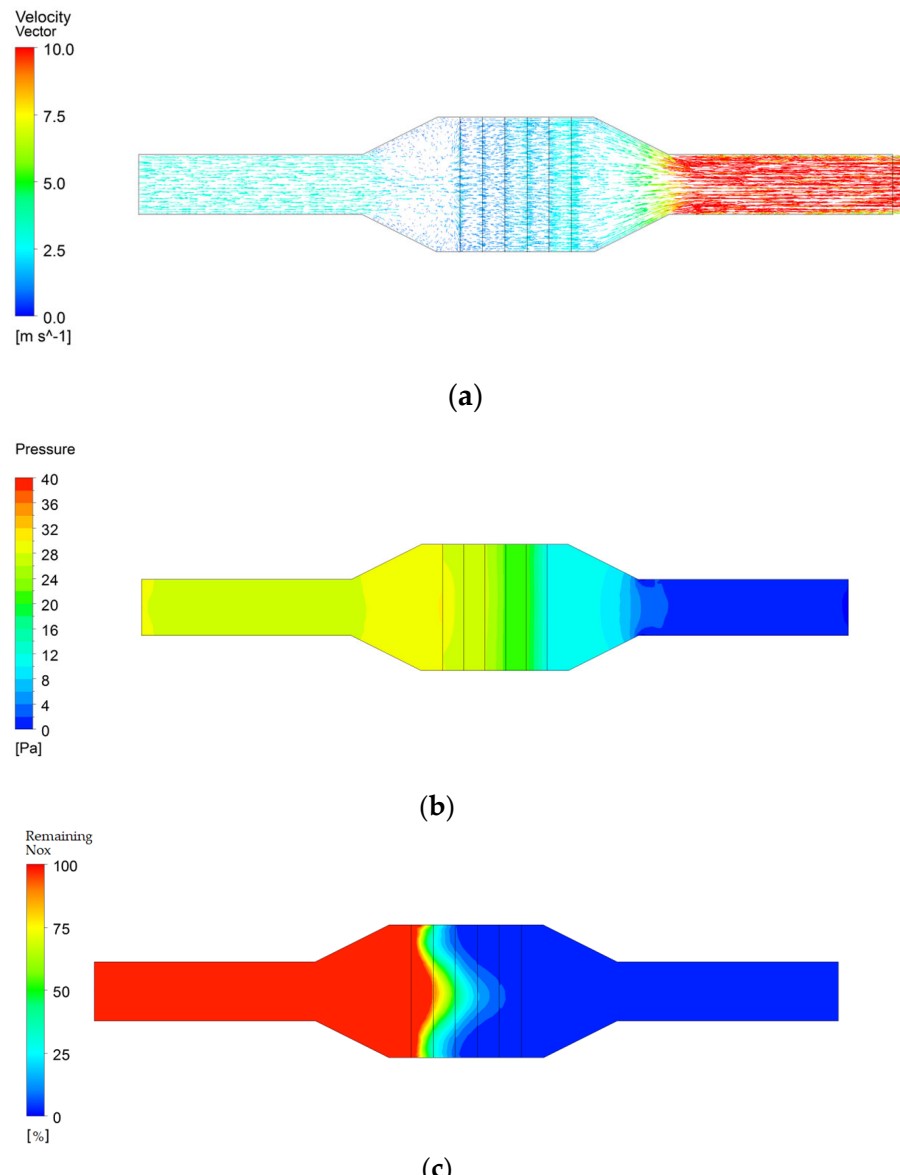

**Figure 8.** NOx reduction in the SCR system in a 1.5-ton boiler (LNG, 50% load): (**a**) velocity results for a boiler load of 50%; (**b**) pressure results for a boiler load of 50%; (**c**) NOx reduction results for a boiler load of 50%.

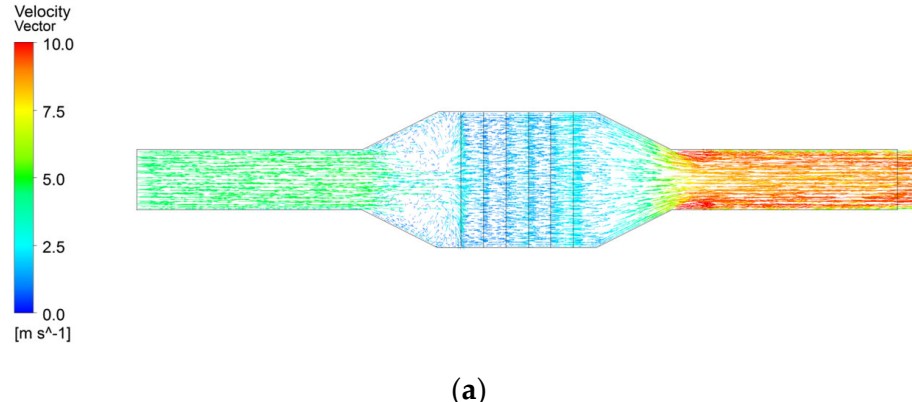

**Figure 9.** *Cont*.

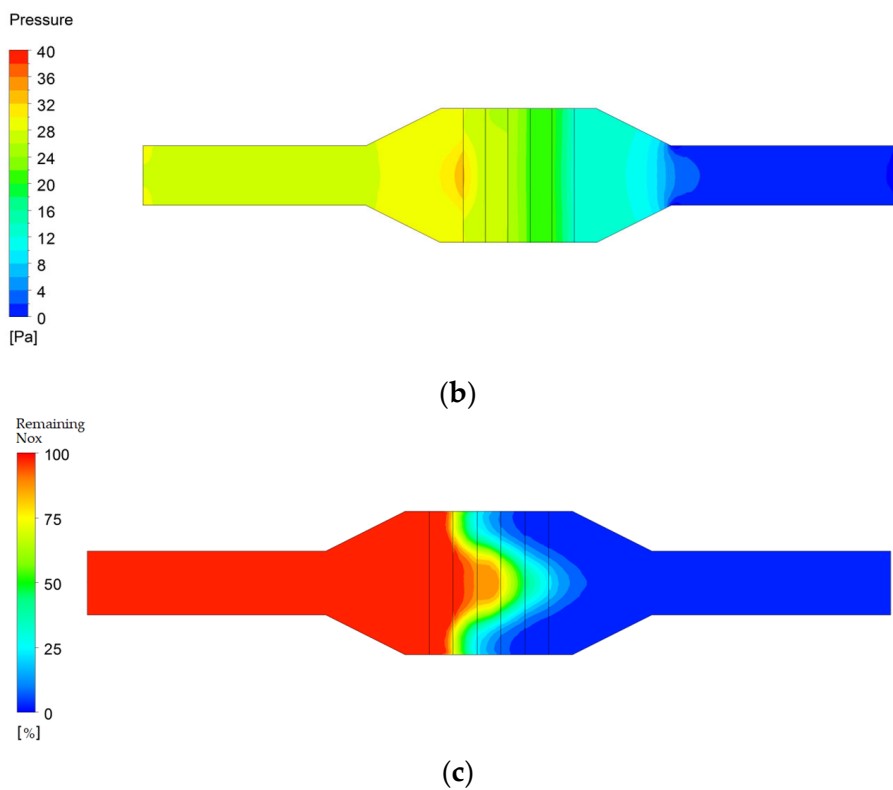

**Figure 9.** NOx reduction in the SCR system in a 1.5-ton boiler (LNG, 75% load): (**a**) velocity results for a boiler load of 75%; (**b**) pressure results for a boiler load of 75%; (**c**) NOx reduction results for a boiler load of 75%.

Figure 10 shows the numerical analysis results when the boiler load using LNG fuel was 100%. Figure 10a shows the flow velocity of the SCR system with an inlet velocity of 2.5–3 m/s and an outlet velocity of 7.5 m/s. Figure 10b shows the pressure difference between the inlet and outlet of the reactor. As a result of checking the pressure change, the pressure difference between the inlet and outlet was 32 Pa when the boiler load was 100%, but the outlet pressure decreased to 0 Pa when the flow passed through the reactor. Figure 10c shows the NOx reduction rate per hour according to the boiler load. The results of the numerical analysis confirmed that NOx was introduced into the inlet at 67 ppm when the boiler load was 100%, but the NOx discharged to the outlet was reduced to 35 ppm. The catalytic reaction was the most activated, and NOx emissions were reduced near the 3/5 point of the inlet of the SCR system reactor.

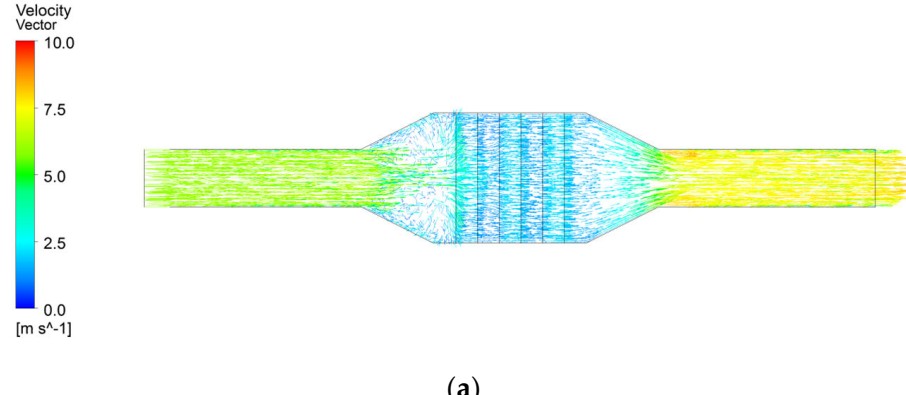

**Figure 10.** *Cont.*

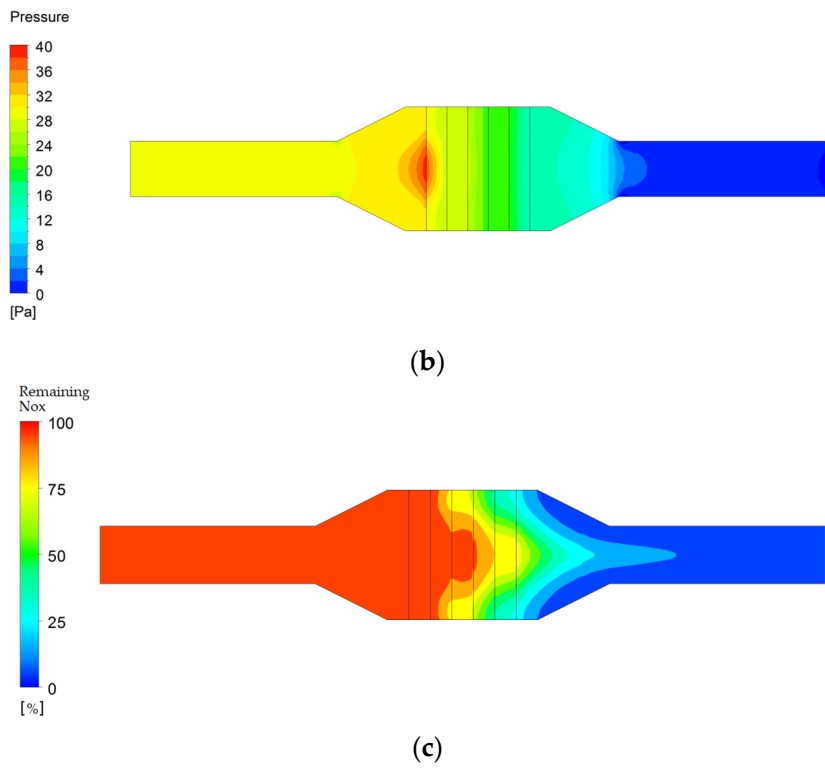

**Figure 10.** NOx reduction in the SCR system in a 1.5-ton boiler (LNG, 100% load): (**a**) velocity results for a boiler load of 100%; (**b**) pressure results for a boiler load of 100%; (**c**) NOx reduction results for a boiler load of 100%.

Figure 11 shows the NOx reduction rate obtained through numerical analysis and experiments with the boiler SCR system when urea was injected at 160 g/h under boiler loads of 50%, 75%, and 100%. When the boiler load was 50%, NOx at the inlet was detected as 67 ppm. The NOx coming out through the outlet was 0 ppm, representing a 100% reduction. As a result of numerical analysis when the boiler load was 75%, the NOx coming out toward the outlet was 23 ppm, indicating a reduction rate of 65.7%. The NOx reduction rate of the experimental value was 24 ppm, indicating a reduction rate of 64.1%. As a result of numerical analysis when the boiler load was 100%, the NOx emissions through the outlet were 34 ppm, which indicated a reduction rate of 49.3%. The NOx reduction rate in the experiment was 35 ppm, which was a 47.8% reduction rate. The difference in the catalytic reaction time between the numerical analysis and the experimental results occurred when the boiler load was 75%. The NOx reduction rate after completion of the catalytic reaction was the same, but the difference in catalytic reaction time was thought to be due to the flow rate. In the case of using LNG fuel, there was a difference in the outlet velocity of the flow compared to the case of using diesel fuel. When using LNG as fuel, the outlet velocity decreased as the load increased. If the outlet velocity was reduced, the retention time of the urea solution increased, increasing the accumulation of urea and reducing the conversion efficiency. In order to overcome this, it was considered that the difference in catalytic reaction time could be reduced if a mixer was installed in the SCR system and the results of the experiment and numerical analysis were compared. When the load was 50% and 100%, the numerical analysis and experimental results were almost identical, with no difference in the catalytic reaction time.

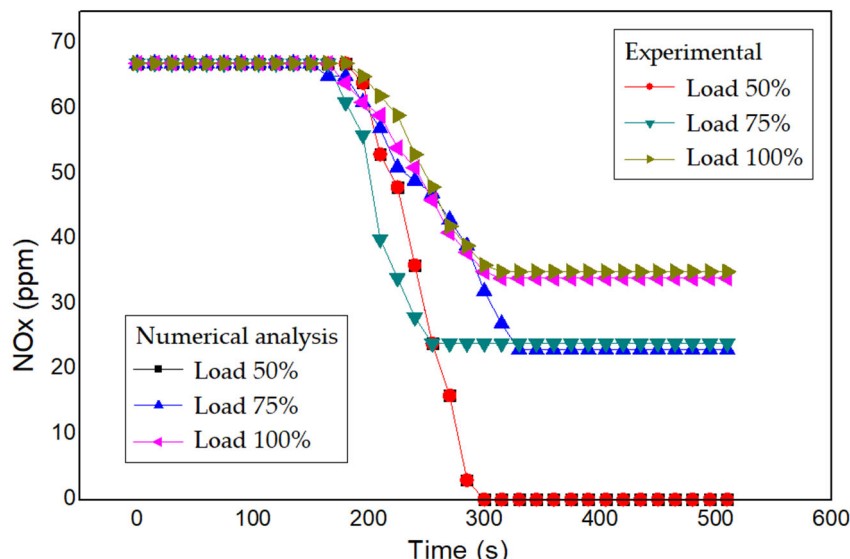

**Figure 11.** NOx emission reduction according to the load when using LNG fuel.

## 4. Conclusions

This study conducted numerical analyses and experiments to examine NOx reduction rates according to 50%, 75%, and 100% loads for a 1.5-ton marine boiler using diesel fuel and LNG fuel. The results are summarized as follows:

(1) The flow velocity of diesel fuel and LNG fuel by boiler load was maintained constant at 2.5–3 m/s. In the case of diesel fuel, the flow velocity through the outlet of the SCR system was constant at 10 m/s. In the case of LNG fuel, when the boiler load was 50%, the outlet velocity was 10 m/s. When the boiler load was 75%, the outlet velocity was 7.5–10 m/s. When the boiler load was 100%, the outlet velocity was 7.5 m/s. Therefore, it was found that as the boiler load increased, the flow velocity through the outlet decreased. Furthermore, the pressure of the SCR system was reduced to 28 Pa at the inlet and to 0 Pa at the outlet when diesel fuel was used. When LNG fuel was used, the inlet pressure was reduced to 32 Pa, and the outlet pressure was reduced equally to 0 Pa.

(2) When the boiler load was 50%, the NOx emissions of diesel fuel and LNG fuel decreased by 100% to 0 ppm. When the load ratio was 75%, the NOx emissions of diesel fuel were reduced by 77.4% to 40 ppm, and for LNG fuel, they were reduced by 64.1% to 24 ppm. At a load ratio of 100%, the NOx emissions of diesel fuel were reduced by 66.1% to 60 ppm, and those of LNG fuel were reduced by 47.8% to 24 ppm. Therefore, it was found that the NOx reduction rate of diesel fuel according to the boiler load was higher than the NOx reduction rate of LNG fuel.

(3) The NOx reduction rate was examined by performing a comparative analysis between the numerical analysis and experiments of the boiler SCR system, according to diesel fuel and LNG fuel under boiler loads of 50%, 75%, and 100%. The urea injection rates were 300 g/h and 160 g/h, respectively. When diesel fuel was used, the difference in catalytic reaction time between the numerical analysis and experimental results occurred when the boiler loads were 50%, 75%, and 100%. When LNG fuel was used, a difference in catalytic reaction time occurred at boiler loads of 75% and 100%. Therefore, it was found that the numerical analysis and experimental results were almost identical. In other words, this study designed an optimal SCR system through numerical analysis, according to the important parameters of the SCR system. Therefore, the findings of this study are applicable to the design of an efficient SCR system for LNG ships. In addition, it was expected that it could be used for research and development to miniaturize the SCR system so that it could be applied to small ships.

In order to develop the SCR system, it was expected that it could be used to predict performance according to fuel and load, if not only experimental methods but also numerical analysis were used.

**Author Contributions:** Conceptualization, J.-U.L. and S.-C.H.; methodology, S.-H.H.; software, J.-U.L.; validation, S.-C.H.; formal analysis, S.-H.H.; investigation, S.-C.H.; resources, J.-U.L.; data curation, S.-C.H.; writing—original draft preparation, J.-U.L.; writing—review and editing, S.-H.H.; visualization, S.-C.H.; supervision, S.-C.H. and S.-H.H.; project administration, S.-H.H.; funding acquisition, S.-H.H. All authors have read and agreed to the published version of the manuscript.

**Funding:** These results were supported by the "Regional Innovation Strategy (RIS)" through the National Research Foundation of Korea (NRF) funded by the Ministry of Education (MOE)(2021RIS-003).

**Institutional Review Board Statement:** Not applicable.

**Informed Consent Statement:** Not applicable.

**Data Availability Statement:** Not applicable.

**Conflicts of Interest:** The authors declare no conflict of interest.

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
