# Peer review of "Numerical and Experimental Study on NOx Reduction According to the Load in the SCR System of a Marine Boiler"

_jmse, doi:10.3390/jmse11040777_

Round 1

Reviewer 1 Report

Dear Editor and Authors,

the submitted article is original and provides new knowledge about NOx reduction using diesel fuel and LNG. I have a few observations that, if corrected, I think the article could be published.
Line 96. m2 should be in superscript.
Line 89. Methodology. I think this section needs to be revised. It is not appropriate to present the data in tables and then describe all the same figures separately in the text. It would be enough to present the values ​​in tables. No need to duplicate the same information.  
Line 89. Methodology. It would be useful if, in this chapter, the measuring equipment used to determine NOx was described separately. Model, measurement limits, measurement resolution, etc.
Line 215 and 292. Y-axis values ​​start at -20 and -10. Meanwhile, the smallest displayed value is 0. NOx cannot be negative in principle, so the y-axis values ​​should also start from 0.
Line 317-318. Isn't the value of 160 and 300 g/hr mixed up here? The methodology (Table 2) states that the injection rate for diesel fuel is 300 g/hr, and for LNG 160 g/hr. Meanwhile, the opposite is true in the conclusions.

Sincerely, Reviewer

Reviewer 2 Report

The manuscript is in requirement of some revision before considering for accepting in the Journal. Some of the points which the authors must adhere to modifying the manuscript are shown below:

1.      The English language needs more enhancement

2.      The highlights need to include and clarify the main findings.

3.      Please, could you rewrite the abstract and conclusion section needs to be more focused on the main goal and the key obtained findings?

4.      Could you add more details for the experimental test rig section? Also, its layout needs more clarification?

5.      There is no conclusion about the recommended blend among those were studied.

6.      What is the authors' perspective about commercially applying this technique in the marine engines? What is the future investigation needed to relate this line of work?

7.       The literature review section needs to rewrite again by updating the recent research.

Author Response

Thank you for your valuable review.
We tried to apply all of your comments.
The detailed modifications are as follows.

1. We will improve through English proofreading in the next process of manuscript submission.

2. The abstract and conclusion sections have been revised throughout to focus on the main goals.

3. The detailed information on the experimental equipment has been modified to make it more readable. Among the experimental equipment, the core technology is the NOx emission measuring device, and the related contents have been modified.

4. The technology in this manuscript is about miniaturization technology to be applied to ship engines. In the future, we plan to use this research to research technology to be mounted on small ships. Related information has been added to the manuscript.

5. Improved the literature review and the overall context of the manuscript.

Reviewer 3 Report

This paper describes the experimental and numerical study on the performance of the selective catalytic reduction system at 50, 75, 100% loads of a 1.5-ton marine boiler with diesel and LNG fuel. The results of different NOx reduction rates for different loads are summarized. However, there are unclear points in the experimental and numerical methods, a detailed description of them should be provided. Major revisions are required for acceptance of the paper. Some questions are listed below.

1.       P. 3 Figure 2

There is no explanation for Figure 2. Please add an explanation of the SCR system used in this study.

2.       P. 4 Figure 3

What is “DCU” in Figure 3? Please add an explanation of it.

3.       P. 4~5 Section 2.2 Numerical Analysis

Please describe in detail the model of the SCR system. Especially, please explain how SCR reactions and urea decomposition reactions are set in the model.

4.       Figure 4, 5, 6, 8, 9, 10 (c)

There is no contour bar. Please add it in the revised manuscript.

5.       P. 9 Figure 7

Please describe why there was a difference in catalytic reaction time between the numerical analysis and experimental results at 50% and 75% load.

6.       P. 12 Figure 11

Please describe why there was a difference in catalytic reaction time between the numerical analysis and experimental results at 75% load.

Reviewer 4 Report

Based on the load situation of a 1.5 ton marine boiler, this paper conducted numerical analysis and experimental research on the performance of the selective catalytic reduction system. Combining the numerical analysis and experimental research results, the selective catalytic reduction system of a marine boiler was systamatically studied. Overall, the paper is highly innovative and logical, and it is recommended to accept it directly.

Reviewer 5 Report

The manuscript investigated the SCR NOx reduction performance on a 1.5-ton marine boiler by experiment and numerical methods. Results were compared by diesel fuel and LNG fuel for different running boiler load. The results may be useful for the design of SCR system for marine boiler and engines. After major revision, it can be accepted for publication.

1.  Numerical method is very import tool in this study, but the platform was not introduced, by commercial software or home-made code?

2. For the injection of urea, which type of agent was used here, urea particle or urea solution? What's temperature of the injection point? How about the decomposition ratio of the urea? Which type of kinetic models were used here for urea-based SCR NOx reduction simulation?

3. The maximum flow values at 50%, 75%, and 100% boiler loads were 750 Am3/hr, 1,125 Am3/hr, and 1,500 Am3/hr. What's this number mean flow rate for flue gas or air supply volume?

4. In Fig.4, Fig.5, Fig. 6, Fig.8, FIg.9 and Fig.10, color bar for (c) of NOx emission are all missing. 

5. In Fig.4, Fig.5, Fig.6, Fig.8, Fig.9 and Fig.10, the (a) velocity. The existing velocity of 10 m/s seems very strange because the inlet velocity is very low 2-3m/s, after SCR reactor how can the velocity increasing so big amount? The gas volume will not change a lot and the temperature will drops, the cross section seems same, how can the velocity increase?

6. In Fig. 7 and Fig.11, what's meaning about the time? The 300-500s is the dynamic time after urea injection or what other meaning? Since the resident time inside of the SCR reactor is very short. Please explain this time.

7. Usually the NOx concentration will be corrected by certain amount of O2 level, what about this data used?

Round 2

Reviewer 3 Report

The paper is well revised according to the comments, it is believed that it is acceptable for publication.

I have one small question.

P. 5, line 246 – 250

Why does the adsorption performance of ammonia decrease as the temperature decreases, although it is reported that the gas is adsorbed more easily as the temperature decreases?

Author Response

Thank you for your valuable review.

The answer to your question is;

It can be seen in the reaction process of the SCR device.
In the SCR reaction, as the reaction temperature increases, the oxidation reaction of ammonia becomes more active and the conversion rate to NO decreases.
As a result, when viewed as a whole, there is a characteristic with maximum activity as the reaction temperature increases.
For this reason, the adsorption performance of ammonia decrease as the temperature decreases.

Reviewer 5 Report

The authors have revised their manuscript according to the first round review comments. I think it can be accepted for publication now.

Author Response

Thank you for your valuable review.
Thank you for spending your valuable time.